


# Land use change affects biogenic silica pool distribution in a subtropical soil toposequence

Dácil Unzué-Belmonte[1], Yolanda Ameijeiras-Mariño[2], Sophie Opfergelt[2], Jean-Thomas Cornelis[3], Lúcia Barão[4], Jean Minella[5], Patrick Meire[1], Eric Struyf[1]

[1]EcosystemManagement Research Group, Department of Biology, University of Antwerp, Universiteitsplein 1C, 2610 Wilrijk, Belgium.
[2]Earth and Life Institute, Environmental Sciences, Université catholique de Louvain, Croix du Sud 2 bte L7.05.10, 1348 Louvain-la-Neuve, Belgium.
[3]Department Biosystem Engineering (BIOSE), Gembloux Agro-Bio Tech (GxABT), University of Liège (ULg), Avenue Maréchal Juin, 27, 5030 Gembloux, Belgium.
[4]ICAAM, Instituto de Ciências Agrárias e Ambientais Mediterrânicas, University of Évora, Apartado 94, 7002-554 Évora, Portugal.
[5]Universidade Federal de Santa Maria (UFSM), Department of Soil Science, 1000 Avenue Roraima, Camobi, CEP 97105-900 Santa Maria, RS, Brazil.

*Correspondence to:* Dácil Unzué-Belmonte (dacil.unzuebelmonte@uantwerpen.be)

**Abstract.** Land use change (deforestation) has several negative consequences for the soil system. It is known also to increase the erosion rate which affects the distribution of elements in soils. In this context, the crucial nutrient Si has received little attention, especially in a tropical context. Therefore we studied the effect of land conversion and erosion intensity on the biogenic silica pools in a subtropical soil in the south of Brazil. Biogenic silica (BSi) was determined using a novel alkaline continuous extraction where Si/Al ratios of the fractions extracted are used to distinguish biogenic silica and other soluble fractions: Si/Al>5 for the biogenic AlkExSi (alkaline extractable Si) and Si/Al<5 for the non-biogenic AlkExSi. Our study shows that deforestation will rapidly (< 50 years) deplete (10- 53%) the biogenic AlkExSi pool in soils . Depletion intensity depends on the slope of the study site. We demonstrate that higher erosion in steeply sloped sites implies increased deposition of biogenic Si in deposition zones near the bottom of the slope, where rapid burial can cause removal of BSi from biologically active zones. Our study highlights the interaction of erosion strength and land use for the BSi redistribution and depletion in a soil toposequence, with strong implications for basin scale Si cycling.

*Keywords*: biogenic silica, alkaline extractable silica, land use change, erosion.

## 1. Introduction

The terrestrial Si-cycle has received strong attention in the past two decades. Multiple studies show its complexity, with a strong interaction between primary lithology and weathering, biotic Si uptake, secondary pedogenic soil Si pools formation and environmental controls such as precipitation, temperature and hydrology (Struyf and Conley, 2012). Lithology controls the primary source of Si through the weathering of silicate minerals of the bedrock (Drever, 1994). This process provides Si to the soil solution in the form of monosilicic acid ($H_4SiO_4$), also referred to as dissolved silicon (DSi). This DSi is absorbed by plants and, upon plant die-off, is resupplied to the soil in the form of relatively soluble (compared to crystalline silicates) biogenic silicates (BSi), usually in the form of phytoliths (plant silica bodies) (Piperno, 2006). Biogenic silica is one of the most soluble forms of Si in soils (e.g. Van Cappellen, 2003), although some pedogenic compounds have similar reactivity as biogenic silica (Sauer et al., 2006; Sommer et al., 2006; Vandevenne et al., 2015a). During soil formation, the DSi release in soil solution through the dissolution of lithogenic and biogenic silicates contributes to the neoformation of pedogenic silicates, i.e. secondary phyllosilicates (Sommer et al., 2006). The biogenic control on the DSi availability in soil increases with weathering degree. Soil mineralogy, strongly governed by geological and climatic conditions, therefore plays a key role in the DSi transfer from soil to plants (Cornelis and Delvaux, 2016). The complex interactions as described above,



controlling the Si cycle in terrestrial ecosystems, are often referred to as the ecosystem Si-filter (Struyf and Conley, 2012), and ultimately determine an important part of the Si fluxes towards rivers.

Land use change is a particularly interesting global change driver to address in this context. Dissolution of soil biogenic silica increases immediately after deforestation (Conley et al., 2008), increasing DSi fluxes out of the soil and the ecosystem. However, in the long-term, Struyf et al. (2010) showed a decrease of overall DSi fluxes from cultivated land. The conversion from forest to croplands decreases the soil biogenic Si stock, the most relevant contributor to the easily available Si pool for plants. The decrease in soil biogenic Si stock has been related to two important factors. The first factor is the harvesting of

crops (Guntzer et al., 2012; Meunier et al., 1999; Vandevenne et al., 2012). Harvest prevents the return of plant phytoliths to the soil, depleting the phytolith pool. The resultant decrease of DSi availability reduces also the formation of non-biogenic secondary Si fractions (Barão et al., 2014). A thorough analysis separating both biogenic and non-biogenic fractions is crucial in this regard, since traditional extraction procedures to quantify biogenic Si may dissolve also non-biogenic Si fractions. The second factor affecting BSi losses is erosion. In cultivated catchments, strong biogenic silica mobilization is

associated to erosion during strong rainfall events (Clymans et al., 2015). During such events, biogenic Si can represent 40% of the total Si inputs to rivers (Smis et al., 2011).

While it is now accepted that cultivation can cause significant changes in soil Si pools and Si fluxes in temperate climates (Keller et al., 2012), the effect of cultivation on (sub)tropical soil Si pools or on soils with volcanic origin is almost unstudied. Only specific cultivated ecosystems, such as rice fields, have been studied (Guntzer et al., 2012) in this regard.

Yet, the increase of firewood, timber, pasture and food crop demand is causing an increase in land conversion to croplands, implying ongoing rapid land degradation in tropical and subtropical forests (Hall et al., 1993). The aim of our study was to investigate the interactive effects of land use change and terrain slope (as a representation of erosion) on the distribution of the BSi pool in a subtropical soil system, derived from a basaltic parent material. For this purpose, we studied terrestrial Si pools in a natural forest and cultivated land, in gently and steeply sloped locations, applying a recently developed alkaline

extraction technique that allows distinguishing between biogenic and non-biogenic alkaline extractable phases.

## 2. Methods

### 2.1. Study area

The study area is situated near Arvorezinha, in the south of Brazil (28°55"54.88" S, 52°6'33.65" W) (Fig. 1). Four sites with identical climatic conditions (warm temperate, fully humid with warm summer, Cfb, (Kottek et al., 2006) were selected.

Annual mean temperature is between 14 and 18°C and annual mean precipitation between 1700 and 1800 mm, (Minella et al., 2014)). The four sampling sites also have the same parent material (rhyodacite). The mineralogy was similar in all sites with sanidine and quartz as main minerals (Ameijeiras-Mariño, 2017). Soil type corresponded to an Acrisol in three of the sites and a Leptosol (IUSS Working Group WRB, 2015) in the fourth one (steep slope of the cropland), with pH values between 4.7 and 5.9. They represent two land uses, a well conserved forest and a cropland, and two slopes (a steep and a

gentle slope), resulting in four different factor combinations (see Fig. 2).

The forest site consists of a semi-deciduous forest with *Araucaria angustifolia*, *Luehea divaricata*, *Nectandra grandiflora* and *Campomanesia guaviroba* as most representative species. Within the same forest area, two adjacent sites with different slopes were chosen, a gentle slope (maximum 10°) and a steeper slope (maximum 18°). In the gentle slope, some scattered small patches of Yerba mate crop (*Ilex paraguarensis*) were recently planted (<3 years ago), occupying less than 10 % of the

study site. All studied sampling locations were separated at least 5 meters from these mate patches.

The cropland sites were located in two geographically separated areas (1.4 km from each other). Deforestation occurred around 50 years ago and experienced the same historical agricultural practices; conventional ploughing and more recently, minimum tillage (Minella et al., 2014). The soil tillage is traditional, based on topsoil mixing and making ridges and furrow,



more or less in level. Crops in the gentle sloped cropland (maximum 7°) rotate between soybean in summer and black oat
(*Avena sativa*) in winter. Some cattle are occasionally feeding during the vegetative stage, however after the oat flourish the
biomass remains to produce mulch (cover) to soybean seeding based on no-till system. The cropland of the steep slope
(maximum 18°) rotates between tobacco or maize in summer and fallow or black oat in winter.

## 2.2. Soil sampling

Bulk soil samples (n=297) were collected during the summer of 2014. In the forest sites, 4 positions along the slope (from
top to bottom) were selected. In the croplands, due to time constraints during the field campaign, only 3 positions along the
slope (from top to bottom) were selected. Three replicate soil pits were dug per position and soil samples were collected
every 10 cm (from top to 50 cm deep) and every 20 cm (from 50 to 110 deep). Deeper depths were sampled every 50 cm
until 200 cm deep or until the saprolite was reached. At each depth, 10 cm of soil (around 2 kg) was collected. At larger
sample intervals, the 10 cm sample was collected in the middle of the depth interval. Soil samples were mixed, dried, gently
crushed and sieved (2 mm) prior analyzes.

Kopecky ring samples were also collected at each sampled depth. Samples were weighted before and after drying at 105°C
in order to calculate bulk densities.

## 2.3. Analysis

One pit per position was selected as a representative pit (FGTR3, FGUMR2, FGLMR2, FGBR1, FSTR2, FSUMR2,
FSLMR2, FSBR2, CGTR1, CGMR3, CGBR3, CSTR3, CSMR1 and CSBR3 (acronyms in Fig. 2)), resulting in a total of 81
samples.

All samples from selected pits (n=81) and also some other extra depths from other pits, in order to confirm the
representativeness of the selected pits, were analyzed for biogenic and non-biogenic Si content, resulting in a total of 145
bulk soil samples (84 on the forest sites and 61 on the croplands). Samples were analyzed in a continuous flow analyzer
(Skalar, Breda, the Netherlands), using a continuous alkaline extraction recently adapted for soils by Barão et al. (2014). The
extraction in 180 mL of 0.5 M NaOH, at 85 °C runs for half an hour. Dissolved Si and dissolved aluminum (Al) are
measured continuously (with the spectrophotometric molybdate blue method and the lumogallion fluorescence method,
respectively), obtaining two dissolution curves which are fitted with first order equations (1).

$$Si_t \ (mg \ g^{-1}) = \left(\sum_{i=1}^{n} AlkExSi_i \times \left(1 - e^{-k_i \times t}\right)\right) + b \times t$$

$$Al_t \ (mg \ g^{-1}) = \left(\sum_{i=1}^{n} \frac{AlkExSi_i}{Si/Al_i} \times \left(1 - e^{-k_i \times t}\right)\right) + \frac{b \times t}{Si/Al_{min}} \tag{1}$$

Where $Si_t$ and $Al_t$ are the concentrations of Si and Al respectively, at any given time. The equations consist on two parts: the
mineral fraction which has a linear dissolution behavior (DeMaster, 1981; Koning et al., 2002) and the fractions exhibiting
non-linear dissolving behavior. For the mineral fraction, the model renders a linear dissolution rate (b) and the Si/Al ratio
($Si/Al_{min}$) of that linear fraction. Non-linearly dissolving fractions are characterized by: the total amount of Si (alkaline
extractable Si (AlkExSi), mg g-1 dry weight of initial sample mass), the Si/Al ratio (concentration of Si over concentration
of Al) of that fraction and its dissolution rate (k, min[-1]). Assuming the same Si and Al release rate from the same compound
and relating the Si and Al concentration equations through the Si/Al ratio, with the three parameters estimated (AlkExSi, k
and Si/Al ratio), the different fractions dissolving non-linearly are differenced. The same model is fitted with one, two or
three first order equations (summation to n in the formula) and the solution showing least error (F test) from the three fits is
kept. The Si/Al ratio of the fraction is used to determine the origin of the non-linear fractions. Barão et al. (2014)
distinguished the following fractions: fractions showing Si/Al ratio >5 was considered as indicative of a biogenic fraction, as
the proportion of Al in phytoliths is low (Bartoli, 1985; Piperno, 2006). A fraction showing a Si/Al ratio <5 was considered
as representative of non-biogenic or pedogenic Si fractions (clay minerals, oxides and organo-Al complexes). We opted to



discard fractions that represent less than 0.1 mg Si g$^{-1}$, as they are smaller or equal to the detection limit of the method

(Barão et al., 2015). Fractions with k<0.1 were discarded as well, as they represent near linearly dissolving fractions.

### 2.3.1. Analysis on selected pits

The following analyses were only carried out on samples from selected pits (n=81):

A portion of the bulk samples was crushed and a sub-sample was heated at 105°C and 1000°C to obtain the dry weight and the losses on ignition (used for checking the quality of the analysis). The total element content was obtained through borate

fusion (Chao and Sanzolone, 1992) of another sub-sample of the crushed sample; 100 mg were melted at 1000°C for 5 min in a graphite crucible with 0.4 g Li-tetraborate and 1.6 g Li-metaborate, then cooled and dissolved in 100 ml of 2M HNO$_3$ under magnetic agitation at 90–100°C. Elemental contents were determined by Inductively Coupled Plasma-Atomic Emission Spectrometry (ICP-AES); the total reserve of bases (TRB=[Ca]+[Mg]+[K]+[Na]) was calculated afterwards. TRB is commonly used as a weathering index as it estimates the content of weatherable minerals (Herbillon, 1986).

Particle size distribution was executed with a Beckman Coulter device (LSTM-13320).

The mineralogy of sand and silt fractions was determined by powder X-ray diffraction (XRD, Cu Ka, D8). Clay fraction mineralogy was assessed by XRD after K$^+$ and Mg$^{2+}$ saturation, ethylene glycol solvation and thermal treatments at 300 and 550°C (Robert and Tessier, 1974).

AlkExSi pools or stocks every 10 cm depth (kg Si m$^{-2}$) for selected pits were calculated according to Eq. (2).

$$AlkExSi\ stock\ (kg\ Si\ m^{-2}) = \frac{[AlkExSi] \times BD \times h}{100} \qquad (2)$$

where [AlkExSi] is the concentration (mg g$^{-1}$) obtained in the alkaline continuous extraction, BD is the bulk density (g cm$^{-3}$) of that sample, h is the thickness of the depth interval of the sample (cm) and 100 is the conversion factor from mg cm$^{-2}$ to kg m$^{-2}$.

This calculation takes into account the bulk density of each sample, correcting the amount of AlkExSi per gram of dried soil

according to the water content at that specific soil depth. It also calculates the amount of AlkExSi in relation to the thickness of the interval collected (10 cm). For larger intervals, where only 10 cm was collected at mid-interval depth, values of the non-sampled depths were linearly interpolated between two known values. The result is given in kg per square meter, in our case, of every 10 cm deep.

In order to estimate the Total biogenic and non-biogenic AlkExSi pools per pit, the sum of 10 cm-depth biogenic and non-

biogenic AlkExSi pools of each pit was made.

Once having the biogenic and the non-biogenic AlkExSi pools per pit, averages between the 3 (for the croplands) or 4 (for the forests) selected pits were made, in order to assign an Averaged biogenic and non-biogenic AlkExSi pool values to the slope and be able to compare AlkExSi pools between different sites. Then, comparisons between the different study sites were made. In order to compare the biogenic and non-biogenic AlkExSi pools from the forests with the croplands, two

different methods were considered, taking into consideration that the number of positions along the slope in the forest sites is higher than in the cropland sites (4 and 3 respectively): Average 1 using all available measurements for the forest (the 4 positions along the slope) and cropland sites, and Average 2, using a pre-calculated average between upper and lower middle positions measurements in the forest sites.

To study the accumulation of biogenic and non-biogenic AlkExSi pools at the bottom of the slope we have calculated the

Accumulation (AC) using the pool in the bottom compared to the summed pools along the slope for the forests (Eq. (3)) and the croplands (Eq. (4)). The closer the AC value is to 100%, the higher the accumulation results.

$$AC_{Forest} = \frac{AlkExSi_{bottom}}{AlkExSi_{top}+AlkExSi_{upper\ middle}+AlkExSi_{lower\ middle}+AlkExSi_{bottom}} \times 100 \qquad (3)$$

$$AC_{Cropland} = \frac{AlkExSi_{St-bottom}}{AlkExSi_{top}+AlkExSi_{middle}+AlkExSi_{bottom}} \times 100 \qquad (4)$$



## 3. Results

### 3.1. Soil physico-chemical characteristics

Results from total element content, particle size, bulk density and TRB values for selected pits are shown in Tables S2-S4. The XRD mineralogical analysis of the bedrock (rhyodacitic volcanic rocks) reveals that sanidine (feldspar group) is the most abundant mineral (45-55%), followed by very fine grained quartz (~38%) embedded in a matrix of hematite, goethite and clays (~8%) (Ameijeiras-Mariño, 2017). Bulk densities of selected pits ranged from 0.7 to 1.54 mg cm$^{-3}$.

### 3.2. AlkExSi concentrations

AlkExSi values (mg g$^{-1}$ dried soil) with the correspondent k values and Si/Al ratio per fraction are presented in the Table S1. In order to distinguish fractions according to the Si/Al ratio, the thresholds applied by Barão et al. (2014) were used: fractions showing Si/Al ratios above 5 were considered to be biogenic and fractions showing Si/Al ratios below 5 were considered to be non-biogenic fractions.

Figure 3 shows the concentrations of biogenic (Si/Al >5) and non-biogenic AlkExSi fractions (Si/Al < 5) within the soil profiles of selected pits. Overall, the highest concentrations of biogenic AlkExSi appear in the top of the profiles or near the surface and decreases with depth and they also are more abundant at the bottom positions of the slopes. On the other hand, non-biogenic AlkExSi fractions are in most cases absent in the top soil layers and increase in concentration with depth.

### 3.3. AlkExSi pools

The biogenic and non-biogenic AlkExSi pools of selected pits every 10 cm are presented in Table 1. The averages of biogenic and non-biogenic AlkExSi pools per position, land use and slope are shown in Table 2.

In order to make comparisons between different sites the averaged biogenic and non-biogenic AlkExSi pools of the whole slope/site were calculated, taking the average of the pools of the 4 (in forests sites) and 3 (in croplands sites) selected pits along the slope. The averages were: $14 \pm 5.0$ kg m$^{-2}$ and $10 \pm 7.6$ kg m$^{-2}$ (biogenic and non-biogenic AlkExSi pool respectively) for the gentle slope of the forest, $17 \pm 8.7$ kg m$^{-2}$ and $20 \pm 10$ kg m$^{-2}$ (biogenic and non-biogenic AlkExSi pool respectively) for the steep slope of the forest, $12 \pm 13$ kg m$^{-2}$ and $13 \pm 3.3$ kg m$^{-2}$ (biogenic and non-biogenic AlkExSi pool respectively) for the gentle slope of the cropland and $8 \pm 6.3$ kg m$^{-2}$ and $2 \pm 1$ kg m$^{-2}$ (biogenic and non-biogenic AlkExSi pool respectively) for the steep slope of the cropland ('Average 1' in Table 2).

As mentioned, a more accurate averaged AlkExSi pools were calculated when comparing forest to cropland ('Average 2' in Table 2). Instead of considering the 4 positions along the slope in the calculation of the Averaged pool for the forest, a pre-calculated average between the upper middle and lower middle position (16.1 kg m$^{-2}$) was used in the calculation, resulting on the following AlkExSi pools averages: $14 \pm 5$ kg m$^{-2}$ and $8.4 \pm 6$ kg m$^{-2}$ (biogenic and non-biogenic AlkExSi pool respectively) for the gentle slope of the forest and $17 \pm 6$ kg m$^{-2}$ and $22 \pm 7$ kg m$^{-2}$ (biogenic and non-biogenic AlkExSi pool respectively) for the steep slope of the forest.

While the gentle and the steep slope of the forest showed near equal biogenic AlkExSi pools (+10%, $14.2 \pm 5$ kg m$^{-2}$ and $16.6 \pm 8.7$ kg m$^{-2}$, gentle and steep slope respectively), non-biogenic AlkExSi pool might be higher (+81%, $10 \pm 7.6$ kg m$^{-2}$ and $20 \pm 10$ kg m$^{-2}$, gentle and steep slope respectively) on the steep slope.

In the cropland, results were slightly different. Both biogenic AlkExSi (+35%, $12 \pm 13$ kg m$^{-2}$ and $7.9 \pm 6.3$ kg m$^{-2}$, gentle and steep slope respectively) and non-biogenic AlkExSi pool (+85%, $13.2 \pm 3.3$ kg m$^{-2}$ and $1.66 \pm 1$ kg m$^{-2}$, gentle and steep slope respectively) were higher on the gently sloped cropland.

When comparing gently sloped forest and cropland ('Average 2' for forests), there was only a negligible difference for biogenic AlkExSi pool (-12%, $13.5 \pm 5$ kg m$^{-2}$ and $12.1 \pm 13$ kg m$^{-2}$, forest and cropland respectively), but non-biogenic AlkExSi might be higher in the cropland (+57%, $8.4 \pm 6$ kg m$^{-2}$ and $13 \pm 3.3$ kg m$^{-2}$, forest and cropland respectively).




On the steep slopes, it was clear that both non-biogenic AlkExSi pool (-90%, 22.1 ± 6.7 kg m$^{-2}$ and 1.66 ± 1 kg m$^{-2}$, forest and cropland respectively ) and biogenic AlkExSi pool (-52%, 16.7 ± 6 kg m$^{-2}$ and 7.92 ± 6.3 kg m$^{-2}$, forest and cropland respectively ) were much lower in the cropland compared to the forest.

Figure 4 shows the biogenic and non-biogenic AlkExSi pools as a soil profile cut from the top to the bottom of the slope, for the four study sites.

The sum of the AlkExSi pools of selected pits per land use and slope, are shown in the Table 2 ('Total (Sum)'). The
accumulation of the biogenic AlkExSi pool at the bottom position of the slope were: 12% for the gentle slope of the forest, 37% for the steep slope of the forest, 15% for the gentle slope of the cropland and 67% for the steep slope of the cropland.

The accumulation of the non-biogenic AlkExSi pool at the bottom position of the slope were: 18% for the gentle slope of the forest, 25% for the steep slope of the forest, 41% for the gentle slope of the cropland and 16% for the steep slope of the cropland.

It is thus clear that a steep slope results in a redistribution of biogenic AlkExSi to the bottom of the slope, but that a similar distribution is not apparent for the non-biogenic AlkExSi pool.

### 3.4. Result summary

One of the most striking observations in our study is the interaction between slope and land use effect. On the steep slope, there is a strong lowering of AlkExSi pools from forest to cropland. In contrast, for the gentle slopes, the observation was
different, with an increase in the non-biogenic AlkExSi pool in the cropland, but similar biogenic AlkExSi pools. It is also clear that there is a strong redistribution of mostly biogenic AlkExSi to the bottom positions of the slope on steeply sloped croplands and forests.

## 4. Discussion

### 4.1. Redistribution of AlkExSi concentrations along the toposequence

In general, the distribution of biogenic AlkExSi shows the same pattern within each pit: the concentration decreases with depth. Highest concentrations were mostly found at the bottom of the slope in every site (with the exception of the gentle slope of the cropland). This agrees with earlier observations on the distribution of biogenic silica along a toposequence in several soil catenas from temperate areas (Sommer et al., 2006). The distribution of non-biogenic AlkExSi shows a complementary pattern. Non-biogenic AlkExSi fractions are rarely present at the top of the profiles but larger concentrations
are found in deeper layers. Similar patterns were reported in a study carried out in arkosic sediment soils in California (Kendrick and Graham, 2004) and for temperate Luvisols in Belgium and Sweden (Barão et al., 2014; Vandevenne et al., 2015a). Upon leaching of DSi after BSi dissolution, the DSi infiltrates and reacts to form e.g. secondary clays. It can also be absorbed onto oxides. The rate of absorption of DSi by oxides is determined by water infiltration rate, pH, water residence time and weathering intensity (Cornelis et al., 2011; Jones and Handreck, 1963). A large amount of oxides in soil (see
'Mineralogy' in Table S4), high DSi supply, strong water infiltration rates and high pH may result in larger concentrations of Si absorbed by oxides. Our studied sites satisfy these conditions with the exception of the pH (4.7-5.9). Uehara and Gillman (1981) suggested that weathered soil systems can result in a desilicated soil enriched in Fe and Al oxides, with pH close to neutral values. Similar processes might occur in our soils, although soils are not desilicated, but do show high weathering intensity.

Biogenic Si concentrations in Vandevenne et al. (2015a) in temperate Luvisols were one order of magnitude lower than in our study. High silica content of the rhyodacite bedrock in our study sites, together with high weathering rates characteristic of tropical and subtropical soils (Drever, 1994), supply a large amount of DSi to the soil. In addition, weathering stimulated by plants is particularly strong in the tropics (Blecker et al., 2006; Kelly et al., 1998); turnover rates of nutrients are also



higher in tropical and subtropical ecosystems than in temperate regions (Alexandre et al., 1997; Derry et al., 2005), due to
high water availability and temperature. Meunier et al. (2010) showed that the DSi supply from the dissolution of basalts was
1.8 times higher than the DSi product from the dissolution of the litter in a Leptosol of La Réunion Island (Indian Ocean).

**4.2. Effects of erosion and land use change on the biogenic AlkExSi pool along the toposequence**

**4.2.1. Erosion**

*Croplands*

For cropland, it is well documented that the harvest of crops exports large amounts of BSi from the system. This generates
BSi depleted systems in the long-term (e.g. Vandevenne et al. (2015b)). Results from Clymans et al. (2011) in long-term
croplands from Sweden showed BSi pool 4 times lower (<3 kg Si m$^{-2}$) than our results (12 kg Si m$^{-2}$).

It is interesting to notice that a redistribution of biogenic AlkExSi occurs along the slope (Fig. 4). A higher slope degree, and
thus higher erosion rate, provokes the loss of material through water erosion and tillage (Govers et al., 1996), transporting
material down slope and resulting in an accumulation of the biogenic AlkExSi pool at the bottom of the slope. In the gently
sloped sample site, biogenic AlkExSi is more stable at the higher positions along the slope while in the steep slope it
accumulates at the bottom.

Guntzer et al. (2012) showed the importance of crop rotation in the turnover and accumulation of phytoliths in soil. The
accumulation of phytoliths is also influenced by the geochemical stability of phytoliths (Song et al., 2012). However, the
crops rotating in both fields are different and have different Si-demand. Maize and black oat are known to have high Si
content, while tobacco and soy have not (Currie and Perry, 2007; Piperno, 2006). The turnover between maize/tobacco and
fallow/black oat on the steep slope might be the reason of the smaller biogenic AlkExSi pool at this site. Moreover, a higher
erosion rate increases the biogenic AlkExSi deposition at the bottom of the steeply sloped cropland. In fact, the TRB in this
slope was higher than at any of the other sites, (the lower the TRB, the more weathered the soil is and on the contrary, the
higher the TRB, the closer to the weathering state of the bedrock the soil is) suggesting that all weathered material has been
already eroded and the saprolite is closer to the surface.

*Forests*

Biogenic AlkexSi pool in the gentle slope of the forest was ~14 kg Si m$^{-2}$. A high phytolith production in this forest,
corresponding to a strong Si uptake by trees and strong internal recycling, can maintain the BSi stock of the soil system.
Equatorial forests from ferrasols in Congo showed a phytolith pool five times smaller than the present results (2.66 kg m$^{-2}$ in
Alexandre et al. (1997)) and in results from Clymans et al. (2011), the amorphous silica pool from temperate forests in
Sweden was close to half our observations (6.7 kg m$^{-2}$).

The biogenic AlkExSi pool was not higher at the bottom position of the gently sloped forest . This suggests that the physical
erosion at this site is low. In the steeply sloped forest, higher erosion rate apparently did provoke the physical loss of
biogenic AlkExSi, potentially decreasing the amount of Si recycled by the vegetation. BSi is consequently transported to the
bottom of the slope before it can dissolve and be recycled by plants, resulting in an accumulation of BSi at the bottom of the
slope (37%). Larger biogenic AlkExSi pools are also found at lower middle position and bottom, which suggest that the
lower middle position at the steep site already receives and accumulates eroded material. This deposition zone could serve as
a location for permanent BSi storage.

**4.2.2. Land use change**

Our results clearly show how land use change can impact the biogenic AlkExSi pool in a subtropical soil system.



The averaged biogenic AlkExSi pool size followed the sequence: ILS > ILG > ARG > ARS. Overall, croplands showed a 10% and 53% lower (for gentle and steep slopes respectively) biogenic AlkExSi pool compared to well-conserved forest. This loss of biogenic AlkExSi has previously been described in other studies. Vandevenne et al. (2015b) showed similar results for temperate Belgian Luvisols, where croplands showed a decrease of total biogenic AlkExSi of 35% compared to the temperate forest. Results from Clymans et al. (2011) support the same pattern, showing smaller AlkExSi pools in cultivated systems in Sweden. The absence of a larger decrease in the gently sloped cropland may indicate that deforestation occurred too recently to see such a decrease, only triggered by harvest. Opfergelt et al. (2010) showed reminiscences of phytoliths from the previous forested system in croplands of Cameroon deforested in the early 50s. However, a strong depletion of >50% is seen at the steep slope of the cropland compared to its forested counterpart. Although it has been shown that an increase of erosion rate occurs after the conversion from forests to croplands (Vanacker et al., 2014) and this may affect both croplands, Montgomery and Brandon (2002) described how the erosion rate depends directly on the slope and stressed the importance of landslides. The consequence is a huge increase of the accumulation of biogenic AlkExSi pool at the bottom of the steep slope of the cropland (67% of the total biogenic AlkExSi pool).

### 4.3. Importance of scales and methods

The present study clearly shows how sensitive subtropical soil silica cycling is to deforestation. The croplands in earlier studies, e.g. Vandevenne et al. (2015b), had usually been cultivated for more than 200 years, and BSi depletion was explained as a result of long-term cultivation. However, the croplands in the present study were deforested 50 years ago, highlighting how fast the biogenic AlkExSi pool can be depleted from the soil system when physical erosion is high.

Our results confirm the importance of using a continuous extraction to determine BSi pools in soils (Barão et al., 2014). The non-biogenic AlkExSi fractions would have been determined as BSi if alkaline extractions applying only analyses during the linear phase of the extraction had been used (adaptations of the DeMaster (1981)). We acknowledge that some difficulties still remain when applying the method we used. The dissolution in NaOH does not show a real reactivity within soils: the non-biogenic AlkExSi fractions probably have lower solubility in soils (Ronchi et al., 2015) or water (Unzué-Belmonte et al., 2016) than BSi. Using the Si/Al ratio thresholds described for temperate soils to determine the character of the fractions in a different soil may arise some concerns. Without physical extraction we cannot assure that fractions showing specific ratios (below 5) correspond to the same pedogenic compounds found in temperate soils. The method is also unable to distinguish, among the Si/Al > 5 fractions, between phytoliths and opal A/CT. Under a silica saturated system, silica can precipitate in amorphous structures called Opal-A, that in further transformations could be transformed into Opal-CT and finally microquartz (Chadwick et al., 1987; Drees et al., 1989). Opal deposits were identified at more than one meter deep layers in temperate pastures (Vandevenne et al., 2015b) and the tropics (Alexandre et al., 1997). Moreover, results from (Saccone et al., 2007) showed that the amounts of easy-soluble silica were larger in deeper horizons, agreeing with the possibility of having Opal-A at deeper layers in our systems.

### 4.4. Effects of erosion and land use change on the non-biogenic AlkExSi pool along the toposequence

#### 4.4.1. Croplands

The averaged total pool of non-biogenic AlkExSi followed the sequence: ILS > ARG > ILG > ARS. With large amounts of BSi exported through the harvest, lower amounts of BSi (easy to dissolve) return to soil (Vandevenne et al., 2012). Due to reduced infiltration of DSi from surficial BSi, a decrease of non-biogenic AlkExSi fractions deeper in the soil could be expected compared to forest. Yet, this does not occur in the gently sloped cropland, where we found a slightly higher averaged pool of non-biogenic AlkExSi than in the forest with similar slope degree. As mentioned before, 50 years of deforestation provided insufficient time to clearly see an effect on the non-biogenic AlkExSi pool as well. A study in Belgian Luvisols under long-term cropland management (Vandevenne et al., 2015a) showed an enlarged non-biogenic AlkExSi pool





in the croplands compared to forest. The authors explained the result by the fact that the high Si-demand from the crops increases the weathering rate from the mineral phases, transforming low-soluble compounds into high-soluble ones (soluble

in NaOH). Studies in rice fields, high Si-demanding crop, showed the importance of Si-oxides phases in the supply of DSi for the plants/crops (Klotzbücher et al., 2016). A combination of a relatively short time period since deforestation, and the increasing demand for Si by the crops compared to forest species, could thus explain the larger non-biogenic AlkExSi pool in gently cropland, compared to forests.

However, the non-biogenic AlkExSi pool of the steeply cropland is almost inexistent. In this case, no compensation from

mineral phases is seen. Dosseto et al. (2011) showed that soil formation in agricultural lands was two orders of magnitude slower than erosion in a study including volcanic soils of Puerto Rico. As well as for the biogenic AlkExSi pool, the high Si-demand by crops together with the higher erosion rate results also in a complete depletion of the non-biogenic AlkExSi pool in the steep sloped cropland.

### 4.4.2. Forests

The steeply sloped forest showed a larger non-biogenic AlkExSi pool, mainly accumulated at top and bottom positions (Fig. 4). It is clear that the continuous long-term biogenic AlkExSi deposition at bottom positions (apparent also at lower middle position) triggers the formation of new non-biogenic AlkExSi phases which corresponds with lower TRB values. Weathering degree has previously been correlated to the amount of pedogenic silica accumulation in sedimentary soils (Kendrick and Graham, 2004) and also, clay minerals and Si absorbed by oxides were reported by Delvaux et al. (1989) and Opfergelt et al.

(2009) respectively, to be largest at most weathered sites in volcanic soils from Cameroon.

### 4.5. Implications

We show how slope and land use change have strong interacting effects on the distribution of the AlkExSi pool in a subtropical soil. In general, our study agrees well with earlier findings in temperate climates: landscape cultivation can diminish soil BSi stocks. While deforestation occurred only 50 years ago, the biogenic AlkExSi pool in the steeply sloped

cropland was only 50% of the pool in steeply sloped forests. In contrast, on the gentle slopes, no similar depletion was observed. This highlights the importance of erosion strength for the rate of depletion. To our knowledge, almost no studies have included slope as a potential factor (Ibrahim and Lal, 2014). It could therefore also be relevant to include erosion rates in studies of BSi in temperate ecosystems.

Silica and carbon cycles are closely related through the production of phytoliths. A recent study showed a positive relation

between soil organic carbon (SOC) and amorphous silica content along a toposequence and along the depth profile (Ibrahim and Lal, 2014). However, a comparison between their and our results is not possible due to the different methods used to extract the silica fractions. The assumed tight relationship between both elements together with the SOC depletion reported of 45% after 11-50 years of conversion from forest to cropland (Wei et al., 2014), hints to similar mechanisms behind both observations. Some studies have indicated that silica could act as a 'carbon protector' through phytoliths formation: carbon is

occluded within the phytoliths and remains stored until phytoliths dissolve (Song et al., 2014). Moreover, some authors suggested that atmospheric carbon sequestration could be enhanced through phytoliths production and subsequent burial (Li et al., 2013; Parr et al., 2010).

Our study highlights the accumulation of biogenic AlkExSi at deposition zones in croplands. Very little is known on the potential Si sink associated with such deposition zones, as little research has actually focused on Si biogeochemistry in these

zones. Deposition of BSi here could be an important sink for Si. As shown earlier in tidal marshes (Struyf et al., 2007), rapid accumulation of BSi can prevent its complete dissolution, resulting in long-term burial and removal from the global biogeochemical Si cycle.




### 5. Acknowledgements

We thank BELSPO for funding the project SOGLO (Soil System under GLObal change, P7/24), all the members of the
SOGLO Project and the University of Santa Maria for their help during field work in Brazil. D. U. B. also thanks the Soil
System Sciences Division of the European Geoscience Union (EGU) for awarding her the best Outstanding Student Poster
Award at the 2014 EGU Assembly.

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





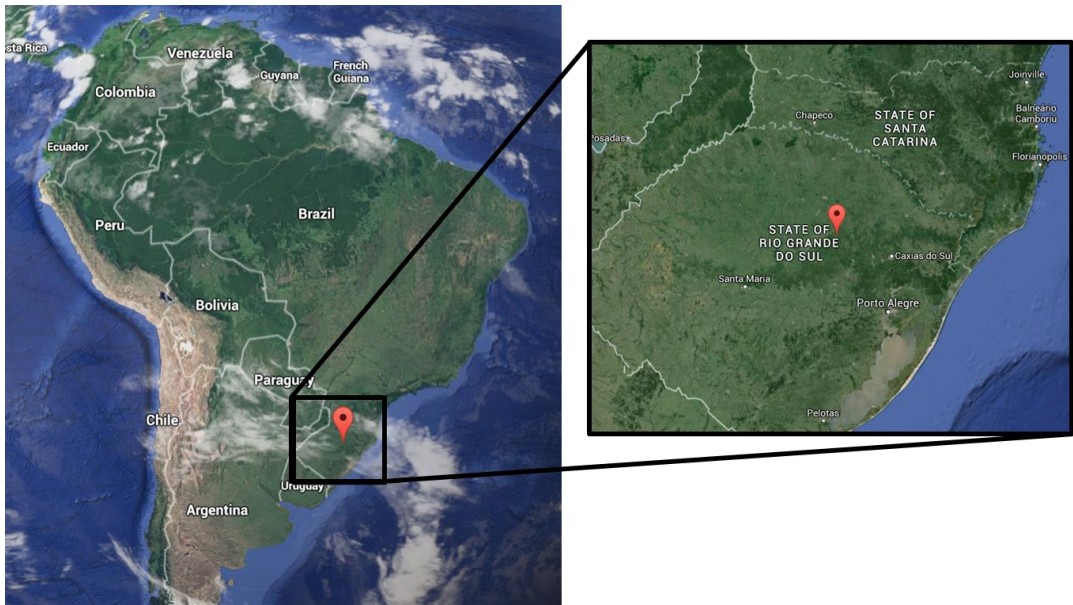

**Figure 1.** Location of study site.

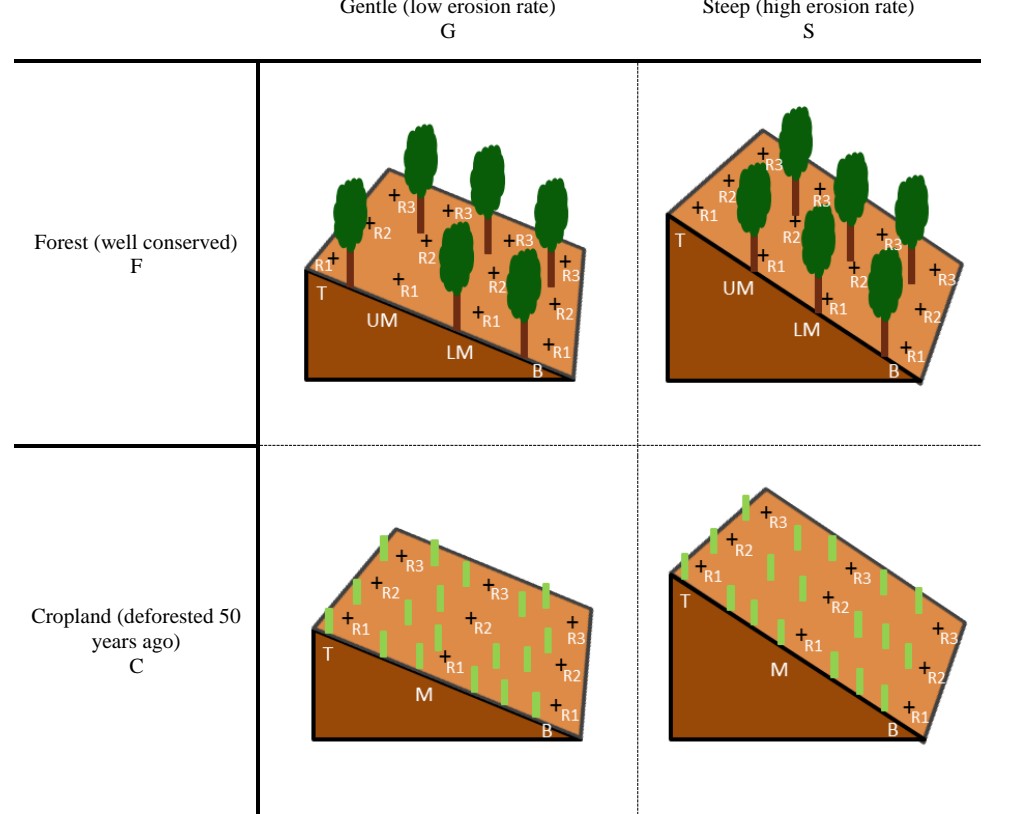

**Figure 2.** Diagram of the studied sites and the acronyms used in the text ordered by ecosystem (F=Forest, C=Cropland), slope (G=Gentle, S=Steep), position (T=Top, UM=Upper middle, LM=Lower middle, M=middle, B=Bottom) and replicate (R1=Replicate 1, R2=Replicate 2, R3=Replicate 3). Plus signs represent sampling points.










**Figure 3.** Biogenic and non-biogenic AlkExSi concentrations (mg g$^{-1}$ dried soil) from selected pits of the sites studied: a) Gentle slope of the forest, b) Steep slope of the forest, c) Gentle slope of the cropland and d) Steep slope of the cropland. Graphs from left to right: Top, upper middle, lower middle (or middle) and bottom pit.






**Table 1.** Biogenic and non-biogenic AlkExSi pools (kg Si m$^{-2}$). $^{!}$pools interpolated

| Position | Depth | Biogenic | Non-biogenic | Position | Depth | Biogenic | Non-biogenic |
|---|---|---|---|---|---|---|---|
| FGTR3 | 0 - 10 | 3.23 | 0.00 | | 130 - 140 | 5.14 | 0.00 |
| | 10 - 20 | 2.84 | 0.00 | | 140 - 150$^{!}$ | 4.07 | 0.61 |
| | 20 - 30 | 1.24 | 0.00 | FSLMR2 | 150 - 160$^{!}$ | 3.02 | 1.20 |
| | 30 - 40 | 2.33 | 0.00 | | 160 - 170$^{!}$ | 1.99 | 1.78 |
| | 40 - 50 | 0.00 | 1.05 | | 170 - 180$^{!}$ | 0.98 | 2.35 |
| | 50 - 55$^{!}$ | 0.81 | 0.24 | | 180 - 190 | 0.00 | 2.90 |
| | 55 - 65 | 2.88 | 0.00 | | 0 - 10 | 2.04 | 0.00 |
| | 65 - 75$^{!}$ | 2.39 | 0.01 | | 10 - 20 | 1.51 | 0.00 |
| | 75 - 85 | 1.90 | 0.02 | | 20 - 30 | 1.57 | 0.00 |
| FGUMR2 | 0 - 10 | 1.63 | 0.00 | | 30 - 40 | 0.74 | 0.00 |
| | 10 - 20 | 0.00 | 1.89 | | 40 - 50 | 0.97 | 0.63 |
| | 20 - 30 | 0.00 | 1.37 | | 50 - 55$^{!}$ | 0.84 | 0.36 |
| | 30 - 40 | 0.00 | 1.35 | | 55 - 65 | 2.38 | 0.83 |
| | 40 - 50 | 0.00 | 1.48 | | 65 - 75$^{!}$ | 2.67 | 0.42 |
| | 50 - 55$^{!}$ | 0.00 | 0.84 | | 75 - 85 | 2.98 | 0.00 |
| | 55 - 65 | 0.00 | 1.81 | FSBR2 | 85 - 95$^{!}$ | 2.51 | 0.77 |
| | 65 - 75$^{!}$ | 0.00 | 0.00 | | 95 - 105 | 2.05 | 1.55 |
| | 75 - 85 | 0.00 | 2.72 | | 105 - 115$^{!}$ | 1.61 | 1.63 |
| | 85 - 95$^{!}$ | 0.94 | 2.24 | | 115 - 125$^{!}$ | 1.15 | 1.71 |
| | 95 - 105 | 2.01 | 1.67 | | 125 - 130$^{!}$ | 0.40 | 0.88 |
| | 105 - 115$^{!}$ | 2.25 | 1.23 | | 130 - 140 | 0.44 | 1.83 |
| | 115 - 125$^{!}$ | 2.49 | 0.80 | | 140 - 150$^{!}$ | 0.36 | 1.86 |
| | 125 - 130$^{!}$ | 1.34 | 0.24 | | 150 - 160$^{!}$ | 0.27 | 1.90 |
| | 130 - 140 | 2.86 | 0.14 | | 160 - 170$^{!}$ | 0.18 | 1.94 |
| | 140 - 150$^{!}$ | 2.31 | 0.38 | | 170 - 180$^{!}$ | 0.09 | 1.97 |
| | 150 - 160$^{!}$ | 1.75 | 0.62 | | 180 - 190 | 0.00 | 2.01 |
| | 160 - 170$^{!}$ | 1.17 | 0.87 | | 0 - 10 | 0.00 | 0.04 |
| | 170 - 180$^{!}$ | 0.59 | 1.12 | | 10 - 20 | 2.74 | 0.02 |
| | 180 - 190 | 0.00 | 1.37 | | 20 - 30 | 0.00 | 4.53 |
| FGLMR2 | 0 - 10 | 1.08 | 0.01 | | 30 - 40 | 3.66 | 0.00 |
| | 10 - 20 | 0.73 | 0.00 | | 40 - 50 | 6.06 | 0.31 |
| | 20 - 30 | 0.00 | 1.19 | | 50 - 55$^{!}$ | 2.78 | 0.09 |
| | 30 - 40 | 0.00 | 0.93 | | 55 - 65 | 5.09 | 0.05 |
| | 40 - 50 | 1.23 | 0.00 | CGTR1 | 65 - 75$^{!}$ | 2.67 | 0.20 |
| | 50 - 55$^{!}$ | 0.71 | 0.00 | | 75 - 85 | 0.00 | 0.36 |
| | 55 - 65 | 1.64 | 0.00 | | 85 - 95$^{!}$ | 0.00 | 1.74 |
| | 65 - 75$^{!}$ | 0.84 | 0.85 | | 95 - 105 | 0.00 | 3.18 |
| | 75 - 85 | 0.00 | 1.74 | | 105 - 115$^{!}$ | 0.83 | 2.29 |
| | 85 - 95$^{!}$ | 1.13 | 0.92 | | 115 - 125$^{!}$ | 1.66 | 1.40 |
| | 95 - 105 | 2.38 | 0.00 | | 125 - 130$^{!}$ | 2.29 | 0.73 |
| | 105 - 115$^{!}$ | 1.75 | 0.55 | | 130 - 140 | 2.92 | 0.05 |
| | 115 - 125$^{!}$ | 1.08 | 1.13 | | 0 - 10 | 0.09 | 0.00 |
| | 125 - 130$^{!}$ | 0.27 | 0.79 | | 10 - 20 | 0.12 | 0.00 |
| | 130 - 140 | 0.00 | 2.06 | CGUMR3 | 20 - 30 | 0.00 | 0.06 |
| FGBR1 | 0 - 10 | 1.60 | 0.00 | | 30 - 40 | 0.00 | 0.45 |
| | 10 - 20 | 1.39 | 0.00 | | 40 - 50 | 0.00 | 4.99 |



| Group | Range | | | Group | Range | | |
|---|---|---|---|---|---|---|---|
| | 20 - 30 | 0.58 | 0.02 | | 50 - 55[1] | 0.00 | 1.59 |
| | 30 - 40 | 1.20 | 0.00 | | 55 - 65 | 0.00 | 1.41 |
| | 40 - 50 | 0.00 | 1.20 | | 0 - 10 | 0.00 | 0.00 |
| | 50 - 55[1] | 0.00 | 0.69 | | 10 - 20 | 0.00 | 2.22 |
| | 55 - 65 | 0.00 | 1.52 | | 20 - 30 | 0.00 | 0.70 |
| | 65 - 75[1] | 0.00 | 1.62 | | 30 - 40 | 0.00 | 1.24 |
| | 75 - 85 | 0.00 | 1.73 | | 40 - 50 | 0.00 | 1.01 |
| | 85 - 95[1] | 0.67 | 0.86 | CGBR3 | 50 - 55[1] | 0.00 | 1.10 |
| | 95 - 105 | 1.34 | 0.00 | | 55 - 65 | 0.00 | 3.38 |
| | 0 - 10 | 0.00 | 2.57 | | 65 - 75[1] | 1.35 | 1.70 |
| | 10 - 20 | 4.08 | 0.00 | | 75 - 85 | 2.69 | 0.02 |
| | 20 - 30 | 0.00 | 4.10 | | 85 - 95[1] | 1.35 | 1.56 |
| | 30 - 40 | 0.00 | 4.42 | | 95 - 105 | 0.00 | 3.09 |
| | 40 - 50 | 0.00 | 3.36 | | 0 - 10 | 1.16 | 0.00 |
| FSTR2 | 50 - 55[1] | 0.00 | 1.87 | | 10 - 20 | 1.26 | 0.00 |
| | 55 - 65 | 0.00 | 4.12 | | 20 - 30 | 0.00 | 0.00 |
| | 65 - 75[1] | 1.21 | 2.13 | | 30 - 40 | 0.00 | 0.17 |
| | 75 - 85 | 2.50 | 0.00 | | 40 - 50 | 0.60 | 0.00 |
| | 85 - 95[1] | 1.27 | 2.80 | CSTR3 | 50 - 55[1] | 0.15 | 0.34 |
| | 95 - 105 | 0.00 | 5.67 | | 55 - 65 | 0.00 | 1.31 |
| | 0 - 10 | 3.49 | 0.00 | | 65 - 75[1] | 0.61 | 0.72 |
| FSUMR2 | 10 - 20 | 3.49 | 0.00 | | 75 - 85 | 1.31 | 0.00 |
| | 20 - 30 | 0.00 | 1.24 | | 85 - 95[1] | 1.24 | 0.24 |
| | 30 - 40 | 0.00 | 2.74 | | 95 - 105 | 1.18 | 0.47 |
| | 0 - 10 | 1.72 | 0.00 | | 0 - 10 | 0.00 | 0.54 |
| | 10 - 20 | 0.00 | 0.49 | CSMR1 | 10 - 20 | 0.00 | 0.39 |
| | 20 - 30 | 0.00 | 1.17 | | 20 - 30 | 0.43 | 0.00 |
| | 30 - 40 | 0.00 | 0.61 | | 0 - 10 | 0.99 | 0.28 |
| | 40 - 50 | 0.00 | 0.80 | | 10 - 20 | 0.74 | 0.00 |
| | 50 - 55[1] | 0.00 | 0.79 | | 20 - 30 | 1.08 | 0.12 |
| | 55 - 65 | 0.00 | 2.31 | | 30 - 40 | 0.80 | 0.00 |
| FSLMR2 | 65 - 75[1] | 0.00 | 1.97 | | 40 - 50 | 1.72 | 0.23 |
| | 75 - 85 | 0.00 | 1.56 | CSBR3 | 50 - 55[1] | 0.89 | 0.05 |
| | 85 - 95[1] | 0.47 | 1.92 | | 55 - 65 | 1.80 | 0.00 |
| | 95 - 105 | 0.94 | 2.28 | | 65 - 75[1] | 1.95 | 0.02 |
| | 105 - 115[1] | 2.09 | 1.66 | | 75 - 85 | 2.11 | 0.05 |
| | 115 - 125[1] | 3.28 | 1.01 | | 85 - 95[1] | 1.95 | 0.03 |
| | 125 - 130[1] | 2.10 | 0.26 | | 95 - 105 | 1.79 | 0.00 |




**Table 2.** Biogenic and non-biogenic AlkExSi pools (kg Si m$^{-2}$), of the selected pits, for the two ecosystems (forest, cropland), for the different slopes (gentle, steep), along different positions along the slope (top, upper middle, lower middle and bottom). Total (sum), Average 1 (Averaged pool between all selected pits) and Average 2 (for the forest sites: Averaged pool between top, pre-calculated average between the upper middle and the lower middle pit (i.e. for the biogenic AlkExSi pool of FG: 16.1 kg Si m$^{-2}$) and bottom pits) of biogenic and non-biogenic AlkExSi pools per site. Accumulation of the biogenic and non-biogenic AlkExSi pools (see Eq. (3) and (4)).

| | Forest | | | | Cropland | | | |
|---|---|---|---|---|---|---|---|---|
| | Gentle | | Steep | | Gentle | | Steep | |
| | Biogenic | Non-biogenic | Biogenic | Non-biogenic | Biogenic | Non-biogenic | Biogenic | Non-biogenic |
| Top | 17.6 | 1.32 | 9.06 | 31.0 | 30.7 | 15.0 | 7.50 | 3.25 |
| Upper middle | 19.3 | 22.1 | 6.98 | 3.98 | | | | |
| Lower middle | 12.9 | 10.3 | 25.8 | 25.7 | 0.21 | 8.50 | 0.43 | 0.93 |
| Bottom | 6.79 | 7.63 | 24.8 | 20.3 | 5.38 | 16.0 | 15.8 | 0.80 |
| Total (Sum) | 56.6 | 41.3 | 66.6 | 81.0 | 36.3 | 39.5 | 23.8 | 4.97 |
| Average 1 | 14.2 ± 5 | 10.3 ± 7.6 | 16.6 ± 8.7 | 20.3 ± 10 | 12.1 ± 13 | 13.2 ± 3.3 | 7.92 ± 6.3 | 1.66 ± 1 |
| Average 2 | 13.5 ± 5 | 8.4 ± 6.1 | 16.7 ± 6 | 22.1 ± 6.7 | | | | |
| Accumulation | 12% | 18% | 37% | 25% | 15% | 41% | 67% | 16% |

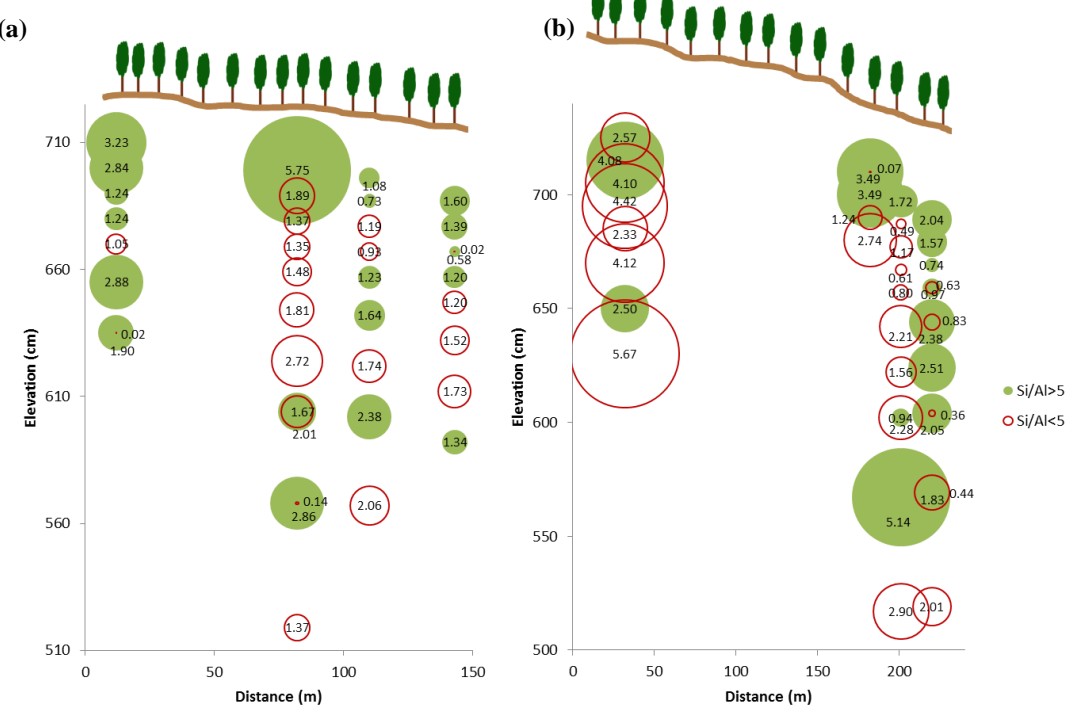




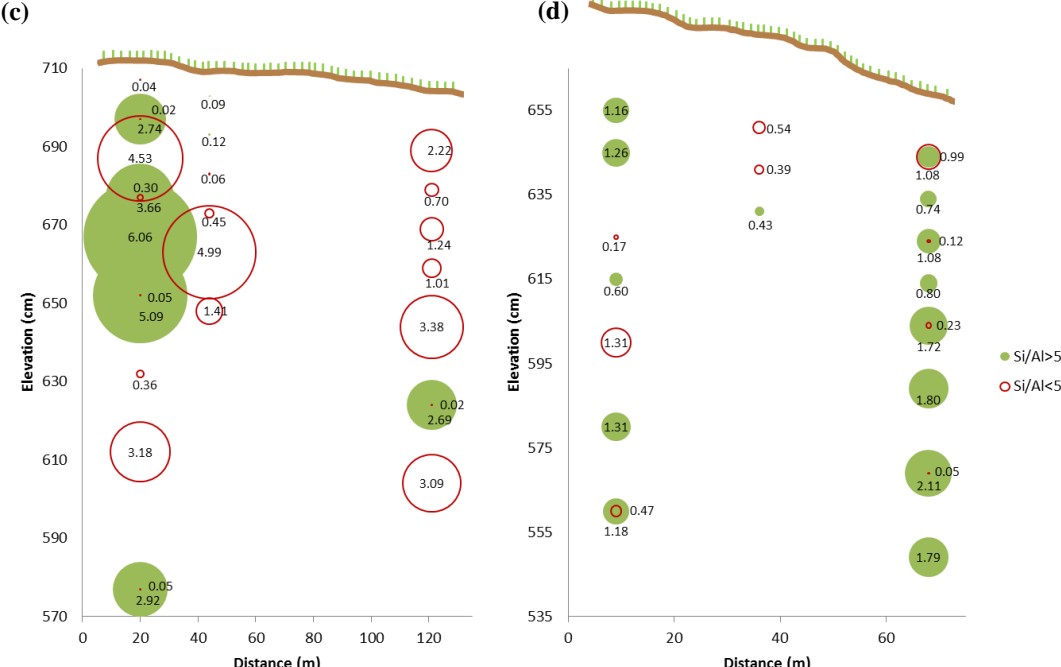

**Figure 4.** Biogenic and non-biogenic AlkExSi pools (kg m$^{-2}$) in the studied sites: a) Gentle slope of the forest (FG), b) steep slope of the forest (FS), c) gentle slope of the cropland (CG) and d) steep slope of the cropland (CS). Green bubbles represent biogenic AlkExSi pools. Red empty bubbles represent non-biogenic AlkExSi pools. Labels show values of the pools (kg m$^{-2}$). Note that the x scales are different.