# Peer review of "Land use change affects biogenic silica pool distribution in a subtropical soil toposequence"

_Solid Earth, 2017_

## Referee Comment (RC1) · Anonymous Referee #1 · 14 Mar 2017

The study of Unzué-Belmonte et al. is very interesting, well written and based on very new techniques. It combines the old story of soil element depletion by erosion and negative effects of deforestation on soil Si pools shown by Struyf et al. 2010. However, despite the manuscript being well written and the data being well discussed I have two major concerns about the manuscript.

major concerns: The first is the complete missing of any statistical analysis. Without statistical analysis it is a descriptive description but nothing is really proven. The second concern is that the authors confound the source of available Si in soils (line 34, line 48, line 383). It is only to minor share the phytoliths which have a high silica condensation state and hence a very slow dissolution. But a much more important source as part of the plant material are the amorphous plant silica deposits like the silicon double layer which has a very low condensation state and is thus highly dissolvable. Phytoliths are more or less stable in nature under common soil conditions. You mentioned it in line 389-391 by indicating phytoliths as a permanent sink for carbon as others found. Phytoliths are not easily dissolvable in soils, otherwise they would not be there for geological scales. Phytoliths are paleo indicators!

minor points: line 14: change to "which negatively affects"

line 116: Sit and Alt should have the same style like in the formula

Table 1 should go for supporting information

line 260: "Highest concentrations were mostly found at the bottom of the slope in every site" this is really no new fact and should not be highlighted that much

Figure 3 and line 379-380 you found no effect of deforestation on biogenic Si pools for gently slop. This is an important outcome contradicting the paper of Struyf et al. 2010. Deforestation is not that bad for soil Si pools, at least for low slopes. This should be highlighted more both in the abstract and conclusion.

---

## Referee Comment (RC2) · Anonymous Referee #2 · 23 Mar 2017

The paper "Land use change affects biogenic silica pool distribution in a subtropical soil toposequence" addresses an important question on which there is so far not much data available, i.e. what changes occur in different Si pools in soil as a consequence of (anthropogenically induced) environmental changes. While the only totally new aspect is the subtropical site studied, the concept is relevant as anthropogenically induced changes in land use and erodibility of soils will, very probably, increase further in subtropical areas. The paper (title, abstract, scope, conclusions) also delivers more or less what it promises: some more work is however needed on the analysis and presentation of the results. The subject is well situated within the scope of the publication.

It is to the authors' credit that they use a fairly novel and under-utilized methodology which makes it easier to differentiate between biogenic and non-biogenic, ecologically and non-geologically speaking potentially relevant Si pools. The description of the

methodology is also (to my, admittedly, fairly well prepared) mind quite adequate and would enable reproduction (with the proper equipment). The field study appears to have been well conducted, although the crops cultivated on the steep and less steep slopes should ideally have been more similar: in this section some more effort to ensure that everyone understands where what samples were taken could have been shown (see specific comments below).

The main weaknesses of the paper are the absence of explicit statistical analyses, the presentation of the results and the discussion of these: the authors could and should make these sections substantially clearer and improve the quality and readability of the paper quite a lot this way. The introduction and most of the methodology sections are quite well written linguistically, but unfortunately the quality of both the text and the structure of the presentation deteriorates towards the end. The results are very likely indeed sufficient to support the interpretations and conclusions, but this should be proven by some kind of statistical analysis (even the use of simple t-tests would improve the paper considerably). These interpretations could also be presented in a clearer fashion (and perhaps the authors would themselves get more out of the results). The citation of existing literature on the subject is adequate (bearing in mind that it is limited).

I think that the paper is well worth publishing as it presents relevant results obtained by appropriate methods and draws fairly good conclusions, but I would recommend that statistical tests are added and quite a lot of the text rewritten.

Specific, technical comments:

The Abstract and Introduction are both well written.

In the Methods-section, it is not quite clear to me what sort of tillage is used (section 2.1 rows 82-).

In section 2.2, please specify how the soils were dried (temperature?). You could

perhaps refer to Fig. 2 here?

In section 2.3, how were the representative pits selected? How were the rest used and why? Please make this clearer! I also hate acronyms and question whether you really need the complicated ones you have designed . . .?

Please also check the overall quality of the language used (starting here); in many instances better words could be used (e.g. not differenced, row 118, smaller THAN or equal to, row 124, nearly linearly dissolving, row 125). Starting from section 2.3.1 the readability of the text also deteriorates somewhat: please consider that this is not a list but a paragraph (e.g. line 127). The paragraph starting on line 151 is, in particular, very raw text (with even capitalization completely haywire) and should be entirely rewritten. Similar specific problems occur throughout the discussion e.g. line 288, 306, 325,328,329, etc.): please have a native speaker check the language once more!

I would also recommend that you re-structure sections 2.3 and 2.3.1 and separate e.g. (laboratory or physicochemical) analyses and calculations into separate sections – it is not that relevant which analyses were made on selected pits and which not. The section on averaging and accumulation is now very hard to follow.

In the Results-section, I would put the supplementary data on AlkExSi-concentrations either together with the bulk data (isn't what is presented in section 3.1 minearology, besides, rather than physico-chemical characteristics?) or after the main results (the pools) as a "footnote". Now it confuses the reader, or at least me: why first concentrations and then pools?

The listing of averages starting on line 182 is not necessary: you already have them in Table 2! Do a general overhaul of the results section and check that what you mention in the text really is necessary and helps the reader to understand what you are talking about, not only a list of numbers.

I would put your result summary into the discussion!

Discussion: section 4.1 is really depth distribution within the soil, isn't it? Could the section header reflect this? And why don't you start with the comparison to other studies now starting at line 240? Likewise, I would begin section 4.2. with a separate paragraph containing the sections on bulk numbers in cropland and forest now within lines 250-253 and 268-272, with or without a separate heading.

The section on crop rotation (line 258-262) is perhaps more land use change than erosion?

Don't you have any references for the effects of erosion on Si pools in forest (line 273-)? Section 4.3. is relevant (but check the language) but would fit better into the discussion later, before the Implications. These are OK; I would perhaps also mention the changes in the non-biogenic pools.

I like your figures, but please ensure that all numbers are of readable size! Especially in Fig. 4 with green background the numbers could be clearer.

―――――――――――――――

---

## Short Comment (SC1) · 14 Apr 2017

In this article, the authors question the impact of land use change in tropical environment, from forested to cultivated land, on the biogenic silica content of soil. They find that in addition to the known impact of harvest (which prevent plants Si to be returned to the soil), erosion also plays a role by moving the superficial, biogenic silica-rich, soil layer downslope where it can be buried. The authors find that increased erosion leads to higher biogenic silica mobilization, eventually leading to stronger biogenic silica depletion in soil. Deforestation and steeper slopes are found to be aggravating factors for this biogenic silica depletion.

The study is globally sound and of good quality and is worth publishing. I do however have some minor concerns detailed below.

[Figure]

1° Erosion vs land-use: The land use change aspect could be discussed more thoroughly. The authors says several times (e.g., L281 and 296) that the impact of deforestation is clear and that the "study clearly shows how sensitive subtropical soil silica cycling is to deforestation". I don't think the impact of land use change is that clear though, and the authors somehow acknowledge it later in the text: "The absence of a larger decrease in the gently sloped cropland may indicate that deforestation occurred too recently to see such a decrease, [. . .]." The authors should be more moderated and discuss more in detail the limited difference of average biogenic AlkExSi content between forest and cultivated sites (gentle slope). They should also discuss more the change of the biogenic AlkExSi distribution within the profiles between the two land uses.

Regarding the erosion aspect, it would be useful to give somewhere in the introduction some data on the erosion fluxes after deforestation. If such data are available in the literature, that would help the reader to get an idea of the importance of the phenomenon at a global or regional scale. Is it just an epiphenomenon or potentially a major Si sink?

2° The authors used an innovative technique to estimate the biogenic silica content in soils. This technique allows differentiating between Si originating from biogenic silica dissolution and from soil minerals during a leaching. The authors discuss the biogenic alkaline extractible Si (AlkExSi) content in the soil profiles, but also spend a lot of time presenting and discussing the non-biogenic AlkExSi content in soil profiles. Non-biogenic AlkExSi can somehow be seen as a proxy for geochemical and mineralogical change in the soil under anthropic pressure, which is interesting, but here the authors spend nearly as much time presenting and discussing the non-biogenic AlkExSi content in soil profiles as they do for biogenic AlkExSi, the object of the study, . . . to in fine say that there is no clear trend to observe. I do understand that a negative result can also be interesting, but in this case the part dedicated to non-biogenic AlkExSi could, to my opinion, be shortened. The potential interest of looking at non-biogenic AlkExSi data should also be clearly explained earlier in the text.

3° Regarding the writing, although the structure is globally good, the phrasing is sometime confusing and the manuscript would greatly benefit from some additional careful readings and reworking to improve the clarity. The Results section for example could be expurgated from long data descriptions that just repeat the content of the tables. Other examples are given in the specific comments below.

Specific comments:

L 22: "that deforestation will rapidly deplete" should be "that deforestation rapidly depleted" as one cannot generalize that easily the observations made here.

L 22 and 283: 10-53 % -> What do these percents correspond to? Is this the remaining fraction of the initial biogenic AlkExSi pool? It's not clear. This problem is recurrent in the text.

L48: change "most relevant" into "most"

L55: "...can represent up to 40%..."

L 99-100: This list of pits is indigestible. Please just mark the pits on figure 2.

L 129: To check the quality of what analysis?

L 184 - 188: Again, this list is painful to read and the data are already in table 2 anyway. Please remove. Also, the number of digits after the comma varies for a same average in the text and in table 2 (E.g., $14\pm5.0$; $14\pm5$; $14.2\pm5$). Please homogenize throughout the manuscript at the correct precision level.

L 189: I don't think these recalculated Si pools are "more accurate"; it's not a question of accuracy but rather a question of making it comparable to the cropland dataset.

L 195 - 206: Here again, the data are not easy to read and to understand. Instead of repeating again the average data, maybe refer to the table 2 for the average values and just give the difference between the sites with different slopes.

L 233: "adsorbed onto oxides" and "adsorption" not "absorbed onto oxides " and "absorption"

L 240-246: I don't understand the point of this paragraph.

L 277: What does this "37%" mean?

L 282: ILS > ILG > ARG > ARS, what are these acronyms? I could not find where in the text they were explained. S and G stand for steep and gentle, but the other letters?

L 290-294: I don't get the point here. Do the authors mean that the slope is a more important parameter than the land-use regarding the erosion intensity? Also, what are these 67% mentioned? Is it the increase of biogenic AlkExSi resulting from landslides? The phrasing is not clear.

L 339-340: the higher abundance of clay mineral and oxides in more weathered soils is not really a specific feature of Cameroonian basalts, it is nearly a definition of soil weathering. The point the authors are trying to make here is not clear.

L 355: It is worth mentioning that some authors also vividly contest this hypothesis.

Figure 3 and 4: unless I missed something, these two figures tell exactly the same thing. The layout is slightly different and one is in mg.g-1 while the other is in kg.m2. . . and that's it. Is there any reason to keep both? I would also suggest to indicate the acronyms of the pits directly on the figure, to make the comparison with the table easier.

───────────────────

---

## Author Comment (AC1) · 11 May 2017

We would like to strongly thank all referees for their thoughtful comments and their appreciation of the paper. The paper has strongly benefited from the suggestions. All minor revisions and rephrasing were accepted as suggested by the reviewers. The text was also checked by a native English speaker. In this response, we present a detailed overview of our responses to all comments.

Anonymous Referee 1

The study of Unzué-Belmonte et al. is very interesting, well written and based on very new techniques. It combines the old story of soil element depletion by erosion and

negative effects of deforestation on soil Si pools shown by Struyf et al. 2010. However, despite the manuscript being well written and the data being well discussed I have two major concerns about the manuscript. major concerns: The first is the complete missing of any statistical analysis. Without statistical analysis it is a descriptive description but nothing is really proven.

> We now include a statistical comparison of average BSi contents of pits from the same position, and also a comparison between the top and the bottom pit from the same slope (accumulation), all for the biogenic AlkExSi pool. We would like to stress that we opted to study a limited amount of soil pits in detail, rather than studying a larger amount of pits in less depth detail. This way, we can provide first insights in both spatial and depth patterns. This however limits the ability to compare Si pools at certain depths within the profile.

The second concern is that the authors confound the source of available Si in soils (line 34, line 48, line 383). It is only to minor share the phytoliths which have a high silica condensation state and hence a very slow dissolution. But a much more important source as part of the plant material are the amorphous plant silica deposits like the silicon double layer which has a very low condensation state and is thus highly dissolvable. Phytoliths are more or less stable in nature under common soil conditions. You mentioned it in line 389-391 by indicating phytoliths as a permanent sink for carbon as others found. Phytoliths are not easily dissolvable in soils, otherwise they would not be there for geological scales. Phytoliths are paleo indicators!

> Although it is true that flora reconstruction based on phytolith preservation is a common paleoecological tool (Kirchholtes et al., 2015; Rovner, 1971) there are several studies that confirm the higher solubility of phytoliths compared to non-biological solid Si phases in soils (Fraysse et al., 2006; Lindsay, 1979; Ronchi et al., 2015; Sommer et al., 2013). The large study of Piperno (2006) about phytoliths clearly shows how variable and how species dependent phytolith characteristics are. Solubility thus is species dependent (Alexandre et al., 1997; Blecker

et al., 2006; Wilding and Drees, 1974). The solubility of phytoliths is further affected by pH, aluminum or Ca2+ concentration in the soil and parent material (Melzer et al., 2012). Cabanes and Shahack-Gross (2015) showed how phytoliths are only partially dissolved in soils. Fraysse et al. (2009) described two different Si pools within the plant: the phytolith pool and Si complexed with the organic matter from the cell walls. Moreover, different condensation states were found in phytoliths depending on the location within the plant (Schaller et al., 2013) which, as suggested by the referee, can result in different solubilities in soil. Borba-Roschel et al., (2006) showed a selective dissolution of phytoliths with depth from the Cyperaceae family. Alexandre et al. (1997) described how 92

line 116: Sit and Alt should have the same style like in the formula

Changed in Line 128.

Table 1 should go for supporting information

We moved the Table to the Supplementary data.

line 260: "Highest concentrations were mostly found at the bottom of the slope in every site" this is really no new fact and should not be highlighted that much

Rephrased in Line 215. "In general, the distribution of biogenic AlkExSi shows the same pattern within each pit: the concentration decreases with depth and highest concentrations are found at the bottom of the slope (with the exception of the gentle slope of the cropland)".

Figure 3 and line 379-380 you found no effect of deforestation on biogenic Si pools for gently slop. This is an important outcome contradicting the paper of Struyf et al. 2010. Deforestation is not that bad for soil Si pools, at least for low slopes. This should be highlighted more both in the abstract and conclusion.

We rephrased Lines 21-23 from the abstract in order to be more moderate in saying that land use change depletes the biogenic AlkExSi pool in a gentle slope: "Our study shows that deforestation can rapidly (< 50 years) deplete the biogenic AlkExSi pool in soils depending on the slope of the study site (10-53It is true that the difference between the biogenic AlkExSi pool of the gently sloped forest and the gently sloped cropland is almost absent in our results, which indeed is in contrast with results from Struyf et al (2010). However, we also point to the fact that this might be due to the recent deforestation: 50 years might not be enough to see the decrease in the terrestrial biogenic AlkExSi pool. We explicitly mentioned the apparent contrast with Stuyf et al. (2010) in Line 272-275: "Our results are apparently in contrast with results from Struyf et al. (2010) who showed a large reduction in DSi export after deforestation in croplands deforested >250 years ago. Nevertheless, the absence of a larger decrease in the gently sloped cropland may indicate that deforestation occurred too recently to see such a decrease, only triggered by harvest."

We would again like to thank you for providing the opportunity to substantially improve our manuscript, and we hope that our paper, which is the first to combine land use change and erosion in the study of terrestrial biogenic Si in subtropical soils, will be accepted for publication in Solid Earth.

Yours sincerely,

Dácil Unzué-Belmonte

Corresponding author

Alexandre, A., Meunier, J.-D., Colin, F. and Koud, J.-M.: Plant impact on the biogeochemical cycle of silicon and related weathering processes, Geochim. Cosmochim. Acta, 61(3), 677–682, doi:10.1016/S0016-7037(97)00001-X, 1997.

Blecker, S. W., Mcculley, R. L., Chadwick, O. A. and Kelly, E. F.: Biologic cycling of

silica across a grassland bioclimosequence, Global Biogeochem. Cycles, 20(May), 1–11, doi:10.1029/2006GB002690, 2006.

Cabanes, D. and Shahack-Gross, R.: Understanding fossil phytolith preservation: The role of partial dissolution in paleoecology and archaeology, PLoS One, 10(5), doi:10.1371/journal.pone.0125532, 2015.

Fraysse, F., Cantais, F., Pokrovsky, O. S., Schott, J. and Meunier, J. D.: Aqueous reactivity of phytoliths and plant litter: Physico-chemical constraints on terrestrial biogeochemical cycle of silicon, J. Geochemical Explor., 88(1–3), 202–205, doi:10.1016/j.gexplo.2005.08.039, 2006.

Fraysse, F., Pokrovsky, O. S., Schott, J. and Meunier, J.-D.: Surface chemistry and reactivity of plant phytoliths in aqueous solutions, Chem. Geol., 258(3–4), 197–206, doi:10.1016/j.chemgeo.2008.10.003, 2009.

Kirchholtes, R. P. J., van Mourik, J. M. and Johnson, B. R.: Phytoliths as indicators of plant community change: A case study of the reconstruction of the historical extent of the oak savanna in the Willamette Valley Oregon, USA, Catena, 132, 89–96, doi:10.1016/j.catena.2014.11.004, 2015.

Lindsay, W. L.: Chemical equilibria in soils, Wiley, New York. [online] Available from: http://soils.ifas.ufl.edu/lqma/SEED/CWR6252/Handout/Chemical equilibira.pdf, 1979.

Melzer, S. E., Cadwick, O. A., Knapp, A. K. and Kelly, E. F.: Lithologic controls on biogenic silica cycling in South African savanna ecosystems, , 317–334, doi:10.1007/s10533-011-9602-2, 2012.

Piperno, D. R.: Phytoliths: A Comprehensive Guide for Archaeologists and Paleoecologists, Altamira Press, San Diego., 2006.

Ronchi, B., Barão, L., Clymans, W., Vandevenne, F., Batelaan, O., Govers, G., Struyf, E. and Dassargues, A.: Factors controlling Si export from soils: A soil column approach, Catena, 133, 85–96, doi:10.1016/j.catena.2015.05.007, 2015.

Rovner, I.: Potential of opal phytoliths for use in paleoecological reconstruction, Quat. Res., 1(3), 343–359, doi:http://dx.doi.org/10.1016/0033-5894(71)90070-6, 1971.

Schaller, J., Brackhage, C., Paasch, S., Brunner, E., Bäucker, E. and Dudel, E. G.: Silica uptake from nanoparticles and silica condensation state in different tissues of Phragmites australis, Sci. Total Environ., 442, 6–9, doi:10.1016/j.scitotenv.2012.10.016, 2013.

Sommer, M., Jochheim, H., Höhn, a., Breuer, J., Zagorski, Z., Busse, J., Barkusky, D., Meier, K., Puppe, D., Wanner, M. and Kaczorek, D.: Si cycling in a forest biogeosystem – the importance of transient state biogenic Si pools, Biogeosciences, 10(7), 4991–5007, doi:10.5194/bg-10-4991-2013, 2013.

Wilding, L. P. and Drees, L. R.: Contributions of forest opal and associated crystalline phases to fine silt and clay fractions of soils, Clays Clay Miner., 22, 295–306, doi:10.1346/CCMN.1974.0220311, 1974.

Please also note the supplement to this comment:
http://www.solid-earth-discuss.net/se-2017-21/se-2017-21-AC1-supplement.pdf

---

## Author Comment (AC2) · 11 May 2017

We would like to strongly thank all referees for their thoughtful comments and their appreciation of the paper. The paper has strongly benefited from the suggestions. All minor revisions and rephrasing were accepted as suggested by the reviewers. The text was also checked by a native English speaker. In this response, we present a detailed overview of our responses to all comments.

Anonymous Referee 2

The paper "Land use change affects biogenic silica pool distribution in a subtropical soil toposequence" addresses an important question on which there is so far not much

data available, i.e. what changes occur in different Si pools in soil as a consequence of (anthropogenically induced) environmental changes. While the only totally new aspect is the subtropical site studied, the concept is relevant as anthropogenically induced changes in land use and erodibility of soils will, very probably, increase further in subtropical areas. The paper (title, abstract, scope, conclusions) also delivers more or less what it promises: some more work is however needed on the analysis and presentation of the results. The subject is well situated within the scope of the publication. It is to the authors' credit that they use a fairly novel and under-utilized methodology which makes it easier to differentiate between biogenic and non-biogenic, ecologically and non-geologically speaking potentially relevant Si pools. The description of the methodology is also (to my, admittedly, fairly well prepared) mind quite adequate and would enable reproduction (with the proper equipment). The field study appears to have been well conducted, although the crops cultivated on the steep and less steep slopes should ideally have been more similar: in this section some more effort to ensure that everyone understands where what samples were taken could have been shown (see specific comments below).

> We tried to clarify in the text where the sampling points were located. The selected pits were marked in Fig. 2 and the ordering of the Result section has been modified in order to improve the comprehensiveness and the flow of the text. We took the suggestions into account and provide more detailed responses below.

The main weaknesses of the paper are the absence of explicit statistical analyses, the presentation of the results and the discussion of these: the authors could and should make these sections substantially clearer and improve the quality and readability of the paper quite a lot this way. The introduction and most of the methodology sections are quite well written linguistically, but unfortunately the quality of both the text and the structure of the presentation deteriorates towards the end. The results are very likely indeed sufficient to support the interpretations and conclusions, but this should be proven by some kind of statistical analysis (even the use of simple t-tests would

improve the paper considerably). These interpretations could also be presented in a clearer fashion (and perhaps the authors would themselves get more out of the results). The citation of existing literature on the subject is adequate (bearing in mind that it is limited). I think that the paper is well worth publishing as it presents relevant results obtained by appropriate methods and draws fairly good conclusions, but I would recommend that statistical tests are added and quite a lot of the text rewritten.

> We answered about the statistics in the response to the previous reviewer: "We now include a statistical comparison of average BSi contents of pits from the same position, and also a comparison between the top and the bottom pit from the same slope (accumulation), all for the biogenic AlkExSi pool.We would like to stress that we opted to study a limited amount of soil pits in detail, rather than studying a larger amount of pits in less depth detail. This way, we can provide first insights in both spatial and depth patterns. This however limits the ability to compare Si pools at certain depths within the profile". We worked on the flow of the writing and had the manuscript reviewed by a native English speaker that made several changes along the whole manuscript.

Specific, technical comments: The Abstract and Introduction are both well written. In the Methods-section, it is not quite clear to me what sort of tillage is used (section 2.1 rows 82-).

> The tillage used in the crop fields was quite intensive since deforestation occurred. More recently a minimum tillage and a cover cropping system were introduced. We adapted the text in Lines 82-90 in order to give a clearer explanation: "Deforestation occurred around 50 years ago and they have since experienced the same historical agricultural practices. Intensive soil tillage occurred from the time of deforestation to 2003, since when a cover cropping and a minimum tillage practice was introduced (Minella et al., 2014). The actual soil tillage is traditional, based on topsoil mixing and making ridges and furrows. Crops in the gently sloping cropland (maximum 7°) rotate between soybean (Glycine max) in summer

and black oat (Avena sativa) in winter. Some cattle occasionally graze during the vegetative stage, and after the oat is harvested the biomass is left to produce mulch (cover) to soybean seeding based on the no-till system. The cropland of the steep slope (maximum 18°) rotates between tobacco (Nicotiana sp.) or maize (Zea mays) in summer and fallow or black oat in winter. The winter crop on this slope is also left behind to produce cover for the next crop".

In section 2.2, please specify how the soils were dried (temperature?). You could perhaps refer to Fig. 2 here?

Included temperature ( 40) in Line 98 and the reference to the Fig. 2 in Line 95.

In section 2.3, how were the representative pits selected? How were the rest used and why? Please make this clearer!

We rephrased Lines 102-104 in order to better explain the selection of the 'representative pits": "One pit per position was selected as a representative pit due to the impossibility of carrying out the novel alkaline extraction analyses on such a high number of samples (297), resulting in a total of 81 samples. The selection avoided pits containing large inclusions (visually) or pits shallower than the other two replicas".

I also hate acronyms and question whether you really need the complicated ones you have designed . . .?

The acronyms used in the study tried to include the information needed to locate the pit in the study schema. Land use type, slope and position in the slope should be given in the acronyms. We opted to change the acronyms for selected pits (removing the replicate number) and marked them in the Fig. 2 as well.

Please also check the overall quality of the language used (starting here); in many instances better words could be used (e.g. not differenced, row 118, smaller THAN or equal to, row 124, nearly linearly dissolving, row 125).

> Changed in Lines 135, 141 and 142 respectively. The manuscript was reviewed by a native English speaker.

Starting from section 2.3.1 the readability of the text also deteriorates somewhat: please consider that this is not a list but a paragraph (e.g. line 127). The paragraph starting on line 151 is, in particular, very raw text (with even capitalization completely haywire) and should be entirely rewritten. Similar specific problems occur throughout the discussion e.g. line 288, 306, 325,328,329, etc.): please have a native speaker check the language once more!

> Changes suggested were made. We had the manuscript reviewed by a native English speaker, who made several changes along the whole manuscript.

I would also recommend that you re-structure sections 2.3 and 2.3.1 and separate e.g. (laboratory or physicochemical) analyses and calculations into separate sections – it is not that relevant which analyses were made on selected pits and which not. The section on averaging and accumulation is now very hard to follow.

> Ordering of the contents in section 2.3 has been changed. Subsections are now as follows: 2.3.1 Physicochemical analyses, 2.3.2. Alkaline continuous extraction and 2.3.3. Post-data treatments.

In the Results-section, I would put the supplementary data on AlkExSi-concentrations either together with the bulk data (isn't what is presented in section 3.1 minearology, besides, rather than physico-chemical characteristics?) or after the main results (the pools) as a "footnote". Now it confuses the reader, or at least me: why first concentrations and then pools?

Concentrations were determined through an extraction and curve modeling afterwards. Pools include interpolated data which makes them an estimation. However, this estimation is needed in order to be able to compare samples to each other, as it includes the humidity of each sample or bulk density.

The listing of averages starting on line 182 is not necessary: you already have them in Table 2! Do a general overhaul of the results section and check that what you mention in the text really is necessary and helps the reader to understand what you are talking about, not only a list of numbers.

List of averages has been removed and Result section has been adapted accordingly. Lines 185-208.

I would put your result summary into the discussion! We changed the location of the Result summary to the beginning of the Discussion. Discussion: section 4.1 is really depth distribution within the soil, isn't it? Could the section header reflect this?

Section header was changed for: "Redistribution of AlkExSi concentrations in depth and along the toposequence".

And why don't you start with the comparison to other studies now starting at line 240? Likewise, I would begin section 4.2. with a separate paragraph containing the sections on bulk numbers in cropland and forest now within lines 250-253 and 268-272, with or without a separate heading.

We clearly separate concentrations from pools in the Results and Discussion. As mentioned before, concentrations are directly measured in the samples and pools include some interpolation. Section 4.1 is about concentrations found in depth and along the toposequence. We compare our results with other comparable results from the literature. Section 4.2 is about the pools. We proceed in the same way comparing our result with the existent published works. Then we discuss the differences found between the different studied sites.

The section on crop rotation (line 258-262) is perhaps more land use change than erosion?

We worked again all over the Discussion section with the objective of improving the readability and the flow of the text. We discarded the headings because we consider that both effects, land use change and erosion, are so interrelated that discussing the two effects separately will distort the message that both together have the strongest impact.

Don't you have any references for the effects of erosion on Si pools in forest (line 273-)?

Studies about Si pools in soils under forest analyzed with a comparable method do not exist as far as we know.

Section 4.3. is relevant (but check the language) but would fit better into the discussion later, before the Implications. These are OK; I would perhaps also mention the changes in the non-biogenic pools.

This study is about the effects on the biogenic AlkExSi pool, which is probably the most effective Si pool that supplies DSi into the soil. Thanks to the method used here we are able to distinguish the biogenic from the non-biogenic AlkExSi. Including a section explaining the benefits of this method (and concerns) is only to demonstrate that all the non-biogenic fractions would have been considered biogenic if other extracting methods were used. Because of that, we give some information about the non-biogenic pool, but it is not the object of study. Limitations exist in characterizing the non-biogenic AlkExSi. The large variability of compounds that might have been dissolved in NaOH and characterized in the fractions with Si/Al<5 make the interpretation of that pool beyond the scope of this manuscript. We would like to remind that the biogenic and non-biogenic AlkExSi fractions described by the method and discussed in the manuscript are only the fractions that showed a first order dissolving behavior. No minerals or

*linearly dissolving fractions are commented in this work. This means that the oxides and clay mineral described in that pool are only part of the total pool of oxides and clays present in the soil. We included Lines 303-305 in order to better introduce the following section about the non-biogenic AlkExSi pool.*

I like your figures, but please ensure that all numbers are of readable size! Especially in Fig. 4 with green background the numbers could be clearer.

*New graphs were made.*

We would again like to thank you for providing the opportunity to substantially improve our manuscript, and we hope that our paper, which is the first to combine land use change and erosion in the study of terrestrial biogenic Si in subtropical soils, will be accepted for publication in Solid Earth.

Yours sincerely,

Dácil Unzué-Belmonte

Corresponding author

Minella, J. P. G., Walling, D. E. and Merten, G. H.: Establishing a sediment budget for a small agricultural catchment in southern Brazil , to support the development of effective sediment management strategies, J. Hydrol., 519, 2189–2201, doi:10.1016/j.jhydrol.2014.10.013, 2014.

Please also note the supplement to this comment:
http://www.solid-earth-discuss.net/se-2017-21/se-2017-21-AC2-supplement.pdf

---

## Author Comment (AC3) · 11 May 2017

We would like to strongly thank all referees for their thoughtful comments and their appreciation of the paper. The paper has strongly benefited from the suggestions. All minor revisions and rephrasing were accepted as suggested by the reviewers. The text was also checked by a native English speaker. In this response, we present a detailed overview of our responses to all comments.

H. Hughes

hhughes@uni-goettingen.de

In this article, the authors question the impact of land use change in tropical environ-

ment, from forested to cultivated land, on the biogenic silica content of soil. They find that in addition to the known impact of harvest (which prevent plants Si to be returned to the soil), erosion also plays a role by moving the superficial, biogenic silica-rich, soil layer downslope where it can be buried. The authors find that increased erosion leads to higher biogenic silica mobilization, eventually leading to stronger biogenic silica depletion in soil. Deforestation and steeper slopes are found to be aggravating factors for this biogenic silica depletion. The study is globally sound and of good quality and is worth publishing. I do however have some minor concerns detailed below. 1 Erosion vs land-use: The land use change aspect could be discussed more thoroughly. The authors says several times (e.g., L281 and 296) that the impact of deforestation is clear and that the "study clearly shows how sensitive subtropical soil silica cycling is to deforestation". I don't think the impact of land use change is that clear though, and the authors somehow acknowledge it later in the text: "The absence of a larger decrease in the gently sloped cropland may indicate that deforestation occurred too recently to see such a decrease, [. . .]." The authors should be more moderated and discuss more in detail the limited difference of average biogenic AlkExSi content between forest and cultivated sites (gentle slope). They should also discuss more the change of the biogenic AlkExSi distribution within the profiles between the two land uses.

We moderated the tone along the manuscript in relation to the differences between the two gently slopes due to the land use change, especially in Lines suggested.

Regarding the erosion aspect, it would be useful to give somewhere in the introduction some data on the erosion fluxes after deforestation. If such data are available in the literature, that would help the reader to get an idea of the importance of the phenomenon at a global or regional scale. Is it just an epiphenomenon or potentially a major Si sink?

We included in Lines 54-57 some information about erosion in agricultural land: "In cultivated catchments, preferential BSi mobilization is associated with erosion

during strong rainfall events (Clymans et al., 2015). During such events, biogenic Si can represent up to 40

2 The authors used an innovative technique to estimate the biogenic silica content in soils. This technique allows differentiating between Si originating from biogenic silica dissolution and from soil minerals during a leaching. The authors discuss the biogenic alkaline extractible Si (AlkExSi) content in the soil profiles, but also spend a lot of time presenting and discussing the non-biogenic AlkExSi content in soil profiles. Nonbiogenic AlkExSi can somehow be seen as a proxy for geochemical and mineralogical change in the soil under anthropic pressure, which is interesting, but here the authors spend nearly as much time presenting and discussing the non-biogenic AlkExSi content in soil profiles as they do for biogenic AlkExSi, the object of the study, . . . to in fine say that there is no clear trend to observe. I do understand that a negative result can also be interesting, but in this case the part dedicated to non-biogenic AlkExSi could, to my opinion, be shortened. The potential interest of looking at non-biogenic AlkExSi data should also be clearly explained earlier in the text.

We reduced the section about the non-biogenic AlkExSi pool and made several changes along the whole manuscript in order to clearly focus the work and results on the biogenic AlkExSi pool. We still discuss partially the results about the non-biogenic AlkExSi pool in order to show that the description of this pool couldn't have been possible if other methods were used.

3 Regarding the writing, although the structure is globally good, the phrasing is sometime confusing and the manuscript would greatly benefit from some additional careful readings and reworking to improve the clarity. The Results section for example could be expurgated from long data descriptions that just repeat the content of the tables.

Several changes were made along the whole manuscript to improve the writing and the long data description in the Results section has been removed.

Other examples are given in the specific comments below. Specific comments: L 22: "that deforestation will rapidly deplete" should be "that deforestation can rapidly deplete" as one cannot generalize that easily the observations made here.

Changed in Line 22.

L 22 and 283: 10-53

The percentages are the differences between the Averaged pools. After adapting the Result section percentages are clearer in Lines 195-203.

L48: change "most relevant" into "most"

Changed in Line 48.

L55: ". . .can represent up to 40

Changed in Line 55.

L 99-100: This list of pits is indigestible. Please just mark the pits on figure 2.

We marked the selected pits in Figure 2 and included the reference to it in the text (Line 104).

L 129: To check the quality of what analysis?

Removed.

L 184 - 188: Again, this list is painful to read and the data are already in table 2 anyway. Please remove. Also, the number of digits after the comma varies for a same average in the text and in table 2 (E.g., 14±5.0; 14±5; 14.2±5). Please homogenize throughout the manuscript at the correct precision level.

Data list was removed and text from Section 3.3 was adapted.

L 189: I don't think these recalculated Si pools are "more accurate"; it's not a question of accuracy but rather a question of making it comparable to the cropland dataset.

Changed in Line 191 for "another average".

L 195 - 206: Here again, the data are not easy to read and to understand. Instead of repeating again the average data, maybe refer to the table 2 for the average values and just give the difference between the sites with different slopes.

Already commented above.

L 233: "adsorbed onto oxides" and "adsorption" not "absorbed onto oxides " and "absorption"

Changed in Lines 222 and 223.

L 240-246: I don't understand the point of this paragraph.

Unfortunately there are not too much data to compare with at this respect. Only few studies have measured biogenic silica pools in soils (mainly in temperate ecosystems) and with comparable methods. The aim of this paragraph is only to relate our results with other existent results and to explain the differences that emerge from that.

L 277: What does this "37perc" mean?

It is the accumulation calculated with Eq. 2 and included in Table 3. We changed in Line 264 the percentage for "(AC of 37perc)".

L 282: ILS > ILG > ARG > ARS, what are these acronyms? I could not find where in the text they were explained. S and G stand for steep and gentle, but the other letters?

Lines 267 and 307 acronyms changed.

L 290-294: I don't get the point here. Do the authors mean that the slope is a more important parameter than the land-use regarding the erosion intensity? Also, what are these 67

> The percentage 67perc is the accumulation calculated (Table 2). We detailed: "(AC of 67perc)" in Line 283.

L 339-340: the higher abundance of clay mineral and oxides in more weathered soils is not really a specific feature of Cameroonian basalts, it is nearly a definition of soil weathering. The point the authors are trying to make here is not clear.

> We rephrased Lines 320-323 in order to clarify that we do not mean that a higher presence of clay mineral and Si oxides is a particular feature of basalts from Cameroon, but that the cited study might be comparable to the rhyodacite bedrock of the sites from this study: "Weathering degree has previously been correlated to the amount of pedogenic silica accumulation in sedimentary soils (Kendrick and Graham, 2004). Further, clay minerals and Si absorbed onto oxides were reported by Delvaux et al. (1989) and Opfergelt et al. (2009) respectively, to be largest at most weathered sites in a study carried out in volcanic soils from Cameroon".

L 355: It is worth mentioning that some authors also vividly contest this hypothesis.

> We included a statement about the existence of different opinions at this respect in Lines 345-348: "Although there are different opinions regarding this topic (Santos and Alexandre, 2017) some have suggested that atmospheric carbon sequestration could be enhanced through phytolith production and subsequent burial (Li et al., 2013; Parr et al., 2010; Song et al., 2016)." , and included the reference:
>
> Santos, G. M. and Alexandre, A.: Earth-Science Reviews The phytolith carbon sequestration concept : Fact or fi ction ? A comment on " Occurrence ,

turnover and carbon sequestration potential of phytoliths in terrestrial ecosystems by Song et al . doi : 10 . 1016 /, Earth Sci. Rev., 164, 251–255.

Figure 3 and 4: unless I missed something, these two figures tell exactly the same thing. The layout is slightly different and one is in mg.g-1 while the other is in kg.m2. . . and that's it. Is there any reason to keep both? I would also suggest to indicate the acronyms of the pits directly on the figure, to make the comparison with the table easier.

> We consider essential the inclusion of the concentrations. AlkExSi concentrations were really measured in the samples. Calculated pools, although needed for comparison, include some interpolation, which make the pools an estimation and not real data. Moreover, we consider that the jump from the raw AlkExSi data directly to pools will be difficult to follow for the readers. We also included the pit acronyms in both Figures.

We would again like to thank you for providing the opportunity to substantially improve our manuscript, and we hope that our paper, which is the first to combine land use change and erosion in the study of terrestrial biogenic Si in subtropical soils, will be accepted for publication in Solid Earth.

Yours sincerely,

Dácil Unzué-Belmonte

Corresponding author

Please also note the supplement to this comment:
http://www.solid-earth-discuss.net/se-2017-21/se-2017-21-AC3-supplement.pdf